# Formal group insertion into aryl C–N bonds through an aromaticity destruction-reconstruction process

Dandan Han[1], Qiuqin He[1] & Renhua Fan[1,2]

Given the abundance and the ready availability of anilines, the selective insertion of atoms into the aryl carbon–nitrogen bonds will be an appealing route for the synthesis of nitrogen-containing aromatic molecules. However, because aryl carbon–nitrogen bonds are particularly inert, anilines are normally activated by conversion to more reactive intermediates such as aryldiazonium salts to achieve functionalization of the aryl carbon–nitrogen bonds, but the nitrogen atom is usually not incorporated into products, instead being discarded. The selective insertion of groups into aryl carbon–nitrogen bonds remains an elusive challenge and an unmet need in reaction design. Here we show an aromaticity destruction-reconstruction process that selectively inserts a trimethylenemethane (TMM) group into the aromatic carbon–nitrogen bond of anilines concomitant with a benzylic carbon–hydrogen bond functionalization. This process provides a transformative mode for anilines, and the insertion products are versatile precursor to various nitrogen-containing aromatic molecules through simple conversions.

[1] Department of Chemistry, Fudan University, 200433 Shanghai, China. [2] Key Laboratory of Functional Small Organic Molecule, Ministry of Education, Jiangxi Normal University, 330022 Nanchang, China. Correspondence and requests for materials should be addressed to R.F. (email: rhfan@fudan.edu.cn)

I n view of the demand for high atom economy, insertion of functional groups into chemical bonds is of significant interest to synthetic chemists. This comes not only from the perspective of fundamental scientific research but also from its potential use in synthetic chemistry. Recently, transition-metal-catalyzed group insertion into unreactive aryl chemical bonds such as carbon–carbon[1–3], carbon–cyanide[4–6], or carbon–halogen[7–9] bonds has drawn particular attention since aromatic molecular complexity can be rapidly built without the generation of stoichiometric amounts of waste products (Fig. 1a). In this context, given the abundance and the ready availability of anilines and their derivatives, the selective insertion of atoms into the aryl carbon–nitrogen bonds will be an appealing route for the synthesis of nitrogen-containing aromatic molecules. However, compared with the cleavage and functionalization of aliphatic carbon–nitrogen bonds[10–18], because aryl carbon–nitrogen bonds are particularly inert, the direct cleavage of these bonds is very difficult[19–23]. Anilines are normally activated by conversion to more reactive intermediates such as aryldiazonium salts[24–27], arylammonium salts[28–30], triazenes[31,32], or amides[33], which serve as electrophiles in various reactions thus forming carbon–heteroatom or carbon–carbon bonds (Fig. 1b). Although these elegant methods have enabled the synthesis of a variety of functionalized aromatic molecules using anilines as the aryl source, the nitrogen atom in substrates is usually not incorporated into products, instead it is being discarded. The selective insertion of groups into aryl carbon–nitrogen bonds remains an elusive challenge and an unmet need in reaction design (Fig. 1c).

Dearomatization of aromatic compounds has been recognized as a fundamental chemical transformation, especially in the synthesis of complex alicyclic molecules[34–40]. The intrinsic functionality and reactivity associated with the aromatic system of anilines may be liberated once the conjugated system is successfully broken up, thus offering a possibility to circumvent the reactivity and the selectivity of anilines[41–49]. In this paper, we report an aromaticity destruction–reconstruction process that selectively inserts a trimethylenemethane (TMM) group into the aryl carbon–nitrogen bond in anilines concomitant with a functionalization of the benzylic carbon–hydrogen bond. This group insertion process provides a transformative mode for anilines and the TMM insertion products are versatile precursors to a variety of nitrogen-containing aromatic molecules through simple conversions.

## Results

**Initial test**. In connection with our recent research on the functionalization of the aryl carbon–nitrogen bonds by using the dearomatization strategy[50], we investigated the reaction of anilines with palladium–TMM (Pd–TMM) complexes under oxidative dearomatization conditions. Pd–TMM complexes in situ generated from 3-acetoxy-2-trimethylsilylmethyl-1-propene and palladium(0) catalysts has served as useful synthons in dipolar cycloaddition with unsaturated bonds in the construction of various cyclic compounds since the first report by Trost in 1979[51–63]. To our delight, in an initial test, we observed the formation of a TMM-containing spiro intermediate and its conversion into compound **3** in which the aryl carbon–nitrogen bond was inserted by the TMM group and the *para*-benzylic carbon–hydrogen bond was functionalized by methoxylation under acidic conditions.

**Optimization of reaction conditions**. Encouraged by these initial results, a set of variables, including palladium catalysts, acidic catalysts, solvents, the ratio of reagents, and temperatures, were screened to establish the optimum reaction conditions (for details, see Supplementary Table 1 in the Supplementary Information). The reaction can be conducted in a one-pot three-step manner. Pd(PPh₃)₄ and Bi(OTf)₃ proved to be the best catalysts for the formation and the conversion of the spiro intermediate, respectively. Moreover, changing the nature of nitrogen protecting group in substrate has a large effect on the transformation. The reaction works with sulfamide groups but not with benzamide or acetamide. Under the optimized conditions, the one-pot reaction of *N*-tosyl protected *p*-toluidine provided the TMM insertion product **3** in 68% yield (Fig. 2).

**Substrate scope**. Substrate scope investigation revealed that the TMM insertion reaction displays broad substrate compatibility (Fig. 3). Under the optimized conditions, the reaction is tolerant of a range of functional groups on the aromatic ring or on the substituents. For example, halogen groups remain unaffected in the palladium-catalyzed reaction leading to the formation of compounds **4** and **5**. Compounds **6** and **7** bearing an allylic or a 1-phenylethyl group were formed in good yields. Reaction of 2-

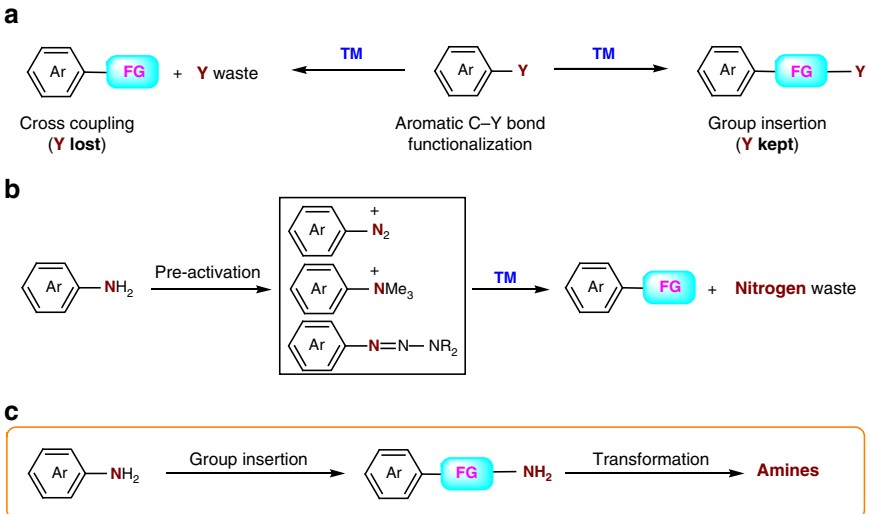

**Fig. 1** Functionalization of aryl C–Y bonds: cross-coupling vs group insertion. **a** functionalization of aromatic C–Y bonds; **b** conventional functionalization of aryl C–N bonds (nitrogen lost); **c** functional group insertion into aryl C–N bonds (nitrogen kept). FG functional group, TM transition metal, Ar aryl group. FG in pink and in a cyan rectangle, TM in blue, and Y, N, nitrogen, NH₂, amines in purple mean emphasis. The different colors used here is for the convenience of reading

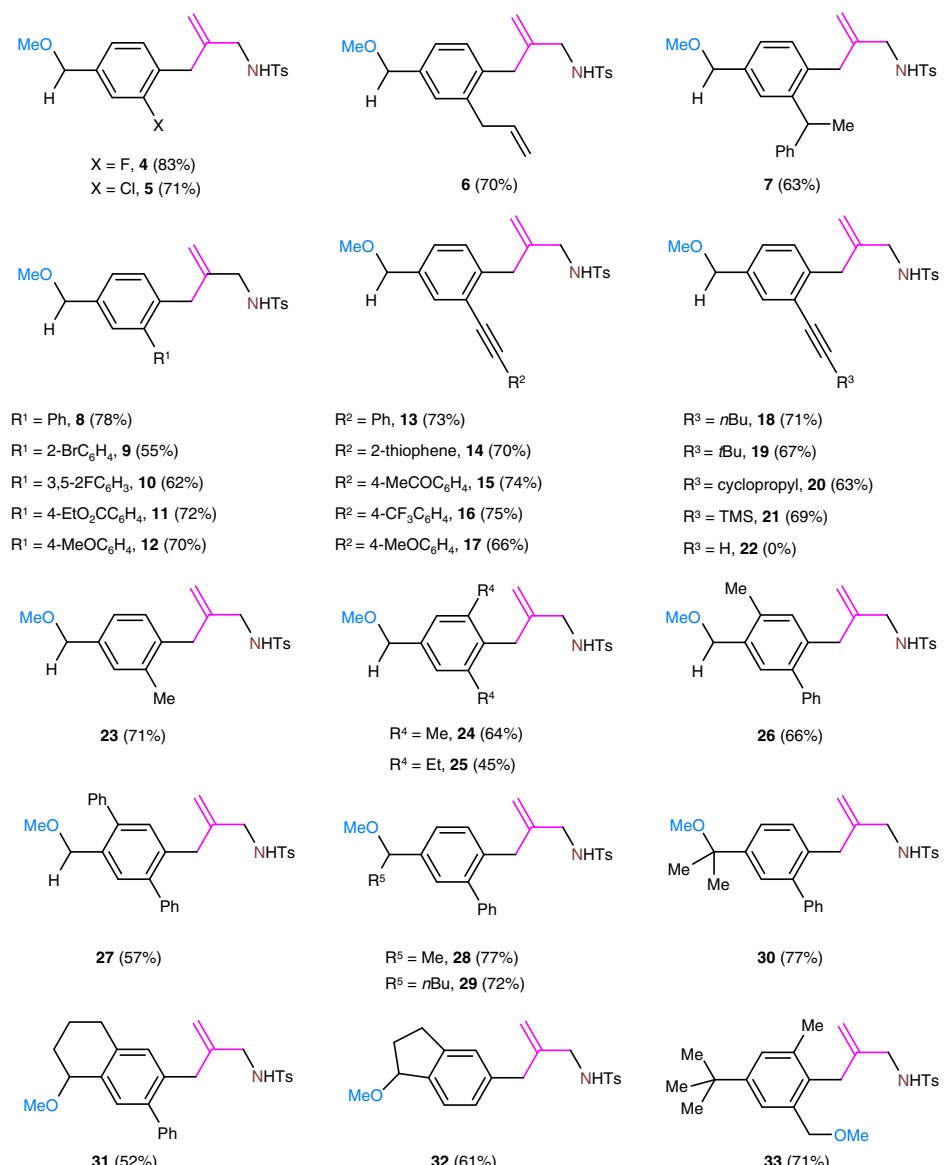

**Fig. 2** TMM insertion into aryl carbon–nitrogen bond in *p*-toluidine concomitant with benzylic methoxylation. Ts tosyl, Ac acetyl, TMS thrimethylsilyl. Trimethylenemethane in pink, which is consistent with the color of FG in Fig.1 means that is insertion group. N in purple is consistent with the color of N in Fig.1. Methoxyl group in cornflower blue means that it is a newly incorporated group

**Fig. 3** Substrate scope investigation. Evaluation of the influence of substituent groups of anilines

aryl- or 2-alkynyl-substituted anilines proceeds smoothly regardless of the different electronic demands on the aryl or the alkynyl substituents. The oxidative dearomatization of substrate bearing an ethynyl group was complex owing to the sensitive of the ethynyl group to the used oxidant. It is noteworthy that, even in the presence of multiple methyl groups at the *ortho* or the *meta* position of the substrate, methoxylation occurs exclusively at the

*para*-benzylic positions. For example, reaction of 2,4-dimethylaniline provided the 4-methoxymethyl-substituted TMM insertion product **23** in 71% yield. Steric hindrance in the TMM insertion reaction was observed, consistent with our hypothesis. For example, compound **25** bearing two *ortho*-ethyl groups was formed in a lower yield than compound **24**, bearing two methyl groups since the *ortho*-ethyl substituents, sterically or otherwise,

**Fig. 4** Plausible reaction pathway. The transformation might proceed via a dearomatization of anilines, an aza-TMM cycloaddition, followed by a subsequent rearomatizing rearrangement and trapping the benzylic carbon cation by nucleophilic attack. PG protecting group. Green scissors mean the bonds there will be cut

**Fig. 5** Variable benzylic functionalization by varying rearrangement solvent. Nuc nucleophile. Nuc and the groups in cornflower blue means that it is a newly incorporated group. Solvent in a cornflower blue square frame means that it acts as a nucleophile

decrease the reactivity of the ketimino group in the dearomatized intermediate. In addition to a methyl group, ethyl, *n*-butyl, or isopropyl groups can be the *para*-substituent of anilines. Tetrahydronaphthalen-2-amine or 2,3-dihydro-1*H*-inden-5-amine are also suitable substrates. Moreover, the TMM insertion process can be extended to substrates lacking a *para*-benzylic C–H bond, and methoxylation takes place in the *ortho*-benzylic position, as in the reaction of 4-(*tert*-butyl)-2,6-dimethylaniline that produces compound **33** in 71% yield. When phenylamine was employed as substrate, the reaction failed to afford the TMM insertion product but gave rise to the corresponding spiro intermediate as the major product.

**Plausible reaction pathway**. Although the precise mechanism of the TMM insertion reaction is not clear at this stage, a plausible pathway could involve the dearomatization of anilines converting the electron-rich aromatic system into an electron-deficient cyclohexadienimine system to permit an aza-TMM cycloaddition of the ketimino group with the Pd–TMM complexes (Fig. 4). This dearomatizing transformation would introduce the TMM group by forming a spiro intermediate. Rearomatizing rearrangement would release the tension of the spiro structure and trapping the benzylic carbon cation by nucleophilic attack would deliver the TMM insertion and benzylic methoxylation products.

**Variable benzylic functionalization**. This assumption led us to investigate the possibility of introducing different functional group into the benzylic position just by varying the solvent in the rearrangement step (Fig. 5). We are delighted to observe the formation of 4-hydroxymethyl-substituted TMM insertion products **34** and **35** when acetone and water was used as a mixed solvent. Moreover, the use of nitriles as solvents instead of methanol led to the formation of the 4-acetamido-substituted products **40–43**.

**Synthetic applications**. To demonstrate the synthetic utility of this TMM insertion process, we explored a number of selective transformations of the insertion products with a view to the synthesis of functionalized nitrogen-containing aromatic molecules (Fig. 6). The representative product **3a** is readily converted to the epoxide **44** by epoxidation to the aziridine **45** by iodocyclization or to the tetrahydroquinoline **46** by reduction and radical amination. The condensation of compound **3a** with allyl bromide followed by an olefin metathesis gave rise to 2,5-dihydro-1*H*-pyrrole **47**[64]. Compound **3a** is also well suited to the construction of 3-azabicyclo[4.1.0]hept-4-ene **48** by reaction with propargyl bromide and subsequent platinum-catalyzed cyclization[65]. The 4-methoxymethyl group in compound **3a** can be

oxidized to a formyl group by treatment with DDQ. The insertion product **24** bearing two *ortho*-methyl groups undergoes reduction and a radical sp[3] C–H amination reaction to form 2,3,4,5-tetrahydro-1*H*-benzo[*c*]azepine **50**[66]. The reactivity of the alkynyl functional group in the insertion products can also be exploited. For example, a gold(I)-catalyzed cyclization of product **13** in the presence of 5 equivalents of $H_2O$ delivers multi-functional benzocycloheptene **51** in 63% yield[67].

## Discussion

In summary, we report an aromaticity destruction–reconstruction process that selectively inserts a TMM group into the aromatic carbon–nitrogen bond in anilines concomitant with a benzylic carbon–hydrogen bond functionalization. The process involves a dearomatization, destroying the aromaticity of anilines, a palladium-catalyzed aza-TMM cycloaddition to introduce the functional group, and a Lewis acid-catalyzed rearrangement to complete the group insertion and restore the aromaticity. The process provides a transformative mode of anilines since the group insertion products are versatile precursors through simple conversions to a range of nitrogen-containing aromatic molecules. Development of an extension of this strategy to other aromatic systems is in progress.

**Fig. 6** Synthetic applications of the TMM insertion products. **a)** *m*-CPBA (6 equiv), CH₂Cl₂, 25 °C, 58%; **b)** NaI (1.2 equiv), *t*-BuOCl (1.2 equiv), MeCN, 25 °C, 50%; **c)** i. H₂ (1 atm), Pd/C, MeOH, 25 °C, 95%, ii. 1,3-diiodo-5,5-dimethylimidazolidine-2,4-dione (1.8 equiv), Na₂SO₃ (2 equiv), ClCH₂CH₂Cl, 60 °C, 83%; **d)** i. 3-bromoprop-1-ene (1.2 equiv), K₂CO₃ (2 equiv), MeCN, 80 °C, 76%, ii. Grubbs catalyst II (4 mol%), CH₂Cl₂, rt, 80%; **e)** i. 3-bromoprop-1-yne (1.2 equiv), K₂CO₃ (2 equiv), MeCN, 80 °C, 80%, ii. PtCl₂ (4 mol%), toluene, 80 °C, 67%; **f)** DDQ (6 equiv), CH₂Cl₂:H₂O = 10:1, 25 °C, 78%. The pink part in the structures mean that it is an insertion group

## Methods

**General method for TMM insertion of anilines**. PhIO (0.11 mmol) was added to a solution of compound **1** (0.1 mmol) in MeOH (2.0 mL) at 25 °C. After 5 min, the reaction mixture was concentrated in vacuo, then was passed through a short silica gel column to remove PhI. The resulting product was mixed with a solution of **2** (0.11 mmol) and Pd(PPh$_3$)$_4$ (0.01 mmol) in anhydrous EtOAc (2.0 mL), and the resulting mixture was stirred at 80 °C for 1 h. Then the reaction mixture was concentrated in vacuo. The resulting crude product was mixed with a solution of Bi (OTf)$_3$ (0.012 mmol) in MeOH (2.0 mL) and stirred at 25 °C for 12 h. After the substrate was consumed completely (monitored by thin-layer chromatographic analysis), the mixture was passed through a short silica gel column and then concentrated under reduced pressure. The residue was purified by flash column chromatography on silica gel (petroleum ether/ethyl acetate = 5/1) to furnish the product **3**. (0.023 mg, 68%). White solid; mp: 80–81 °C; $^1$H NMR (400 MHz, CDCl$_3$): δ 7.70 (d, $J$ = 8.2 Hz, 2 H), 7.28 (d, $J$ = 8.1 Hz, 2 H), 7.22 (d, $J$ = 7.9 Hz, 2 H), 7.06 (d, $J$ = 7.9 Hz, 2 H), 4.99 (s, 1 H), 4.85 (s, 1 H), 4.65 (t, $J$ = 6.3 Hz, 1 H), 4.41 (s, 2 H), 3.45 (d, $J$ = 6.4 Hz, 2 H), 3.38 (s, 3 H), 3.30 (s, 2 H), 2.42 (s, 3 H); $^{13}$C NMR (100 MHz, CDCl$_3$) δ 143.8, 143.4, 137.8, 136.8, 136.3, 129.6, 128.9, 127.9, 127.1, 114.2, 74.4, 58.1, 47.2, 40.0, 21.5; HRMS ($m/z$): [M + H]$^+$ calcd. for C$_{19}$H$_{23}$NO$_3$S, 346.1471; found, 346.1478.

**Data availability**. All data that support the findings of this study are available within this article and its Supplementary Information (including experimental procedures, compound characterization data). Data are also available from the corresponding author upon reasonable request.

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

## Acknowledgements

Financial support from the National Natural Science Foundation of China (21332009, 21572033) is greatly appreciated.

## Author contributions

R.F. directed the research and developed the concept of the reaction with D.H., who also performed the experiments and prepared the Supplementary Methods. Q.H. checked the experimental data. R.F. wrote the manuscript with contributions from the other authors.

## Additional information

**Competing interests:** The authors declare no competing interests.

