## [Peer Review File · Nature Communications]

Reviewer #1 (Remarks to the Author):

The manuscript by Fan and co-workers described an interesting aromaticity destruction-reconstruction process for formal insertion of a trimethylenemethane group into the aromatic C-N bond and concomitant with benzylic C-H bond functionalization. By using this process, a wide range of Ts-substituted anilines could be efficiently converted into the corresponding nitrogen-containing molecules. The present manuscript not only provided a method to synthesis of functional molecules that other method can not be accessed, but also provided a new strategy for cleaving inert C-N bond. Thus, I believe the present manuscript is suitable to be published in Nature Communications after addressing the following points:

- (1) The manuscript is mainly relevant to C-N bond cleavage, thus, the introduction part should be revised and pay more attention to the movement of the C-N activation chemistry. Thus, some reference on the insertion of functional groups such CO, alkene, Carbenoid into C-N bond should be cited.
- (2) The catalyst loading of Palladium used here is 10mol%, is it possible reduce the Pd-loading to 5 mol% or even to 1 mol%?
- (3) The functional group insertion into the C-N bond here is formal insertion, thus, the title should be revised as Formal insertion.....

Reviewer #2 (Remarks to the Author):

This manuscript by Fan and co-worker reported a selective insertion of a trimethylenemethane group into the aryl carbon-nitrogen bond in anilines by a palladium catalyzed aza-TMM cycloaddition of the dearomatized anilines and a subsequent rearrangement reaction. Under the developed conditions, the TMM insertion reactions proceeded well accompanied by the functionalization of the para- or the ortho-benzylic carbon-hydrogen bonds. The key design for the reaction is the realization of an unprecedented group insertion reaction through a simple aromaticity destruction and reconstruction process, which enabled a previously impossible transformation of anilines (the nitrogen atom kept in this reaction vs the nitrogen atom discarded in the previously reported in references 19-33). This reaction also provides a novel application of the dearomatization strategy and the TMM cycloaddition. Moreover, the authors further demonstrated the synthetic utility of the TMM insertion products, and proves that this reaction is not only a new transformative mode of anilines, but also highly useful for the easy conversion of anilines into a variety of nitrogen-containing molecules.

Due to the importance of the reaction and products, this reviewer believes that these results should be highly interesting for a broad range of the readership for Nature Communications. In addition, the manuscript is clearly written, and all the new compounds have been well characterized in the SI. Therefore, publication of this work in Nature Communications is highly recommended.

However, there are some issues need to be addressed as follows:

- 1) How about the reaction involving substrates without an alkyl substituent at the para- or the ortho-position?
- 2) Why did the reaction of substrates bearing an ethynyl group fail to give product 20?

Answers for reviewer 1:**1. Comment:**

The manuscript is mainly relevant to C-N bond cleavage, thus, the introduction part should be revised and pay more attention to the movement of the C-N activation chemistry. Thus, some reference on the insertion of functional groups such CO, alkene, Carbenoid into C-N bond should be cited.

Answer:

Page 2, Introduction, line 10: “However, because aryl carbon-nitrogen bonds are particularly inert, the direct cleavage of these bonds is very difficult” was changed to “However, compared with the cleavage and functionalization of aliphatic carbon-nitrogen bonds¹⁰⁻¹⁸, because aryl carbon-nitrogen bonds are particularly inert, the direct cleavage of these bonds is very difficult”.

References 4-6, 10-12, 14, 17 and 18 in the previous version have been removed.

Eight references about the insertion of functional groups such CO, alkene or carbenoid into the C-N bond have been cited as the references 10-18:

10. Geng, W., Zhang, W.-X., Hao, W. & Xi, Z. Cyclopentadiene-Phosphine/Palladium Catalyzed Cleavage of C-N Bonds in Secondary Amines: Synthesis of Pyrrole and Indole Derivatives from Secondary Amines and Alkenyl or Aryldibromides, *J. Am. Chem. Soc.* **134**, 20230–20233 (2012).
11. Wang, Y., Zhao, F., Chi, Y., Zhang, W.-X. & Xi, Z. Substituent-Controlled Selective Synthesis of N-Acyl 2-Aminothiazoles by Intramolecular Zwitterion-Mediated C–N Bond Cleavage, *J. Org. Chem.* **79**, 11146–11154 (2014).
12. Yu, H., Gao, B., Hu, B. & Huang, H. Charge-Transfer Complex Promoted C–N Bond Activation for Ni-Catalyzed Carbonylation. *Org. Lett.* **19**, 3520–3523 (2017).
13. Yu, H., Zhang, G., Liu, Z.-J. & Huang, H. Palladium-catalyzed carbonylation of allylamines via C–N bond activation leading to β,γ -unsaturated amides. *RSC Adv.* **4**, 64235–54237 (2014).
14. Herzon, S. B. & Hartwig, J. F. Direct, Catalytic Hydroaminoalkylation of Unactivated Olefins with N-Alkyl Arylamines. *J. Am. Chem. Soc.* **129**, 6690–6691 (2007).

15. Miyazaki, Y., Ohta, N., Semba, K. & Nakao, Y. Intramolecular Aminocyanation of Alkenes by Cooperative Palladium/Boron Catalysis. *J. Am. Chem. Soc.* **136**, 3732–3735 (2014).
16. Hu, J., Xie, Y. & Huang, H. Palladium-Catalyzed Insertion of an Allene into an Aminoal: Aminomethylamination of Allenes by C–N Bond Activation. *Angew. Chem. Int. Ed.* **53**, 7272–7276 (2014).
17. Qin, G., Li, L., Li, J. & Huang, H. Palladium-Catalyzed Formal Insertion of Carbenoids into Aminoalcohols via C–N Bond Activation. *J. Am. Chem. Soc.* **137**, 12490–12493 (2015).
18. Harada, S., Kono, M., Nozaki, T., Menjo, Y., Nemoto, T. & Hamada, Y. General Approach to Nitrogen-Bridged Bicyclic Frameworks by Rh-Catalyzed Formal Carbenoid Insertion into an Amide C–N Bond. *J. Org. Chem.* **80**, 10317–10333 (2015).

2. Comment:

The catalyst loading of Palladium used here is 10mol%, is it possible reduce the Pd-loading to 5 mol% or even to 1 mol%?

Answer:

Reactions using 5 mol% or 1 mol% palladium catalysts have been conducted, and the results have added in the Supplementary Table 1 as the entries 38 and 39.

3. Comment:

The functional group insertion into the C-N bond here is formal insertion, thus, the title should be revised as Formal insertion.....

Answer:

The title of the manuscript is changed to: “Formal Group Insertion into Aryl C–N Bonds through An Aromaticity Destruction-Reconstruction Process”.

Answers for reviewer 2:

1. Comment:

How about the reaction involving substrates without an alkyl substituent at the para- or the ortho-position?

Answer:

Page 4, Substrate scope, line 23: “When phenylamine was employed as substrate, the reaction failed to afford the TMM insertion product, but gave rise to the corresponding spiro intermediate as the major product.” was added.

2. Comment:

Why did the reaction of substrates bearing an ethynyl group fail to give product 20?

Answer:

Page 4, Substrate scope, line 8: “The oxidative dearomatization of substrate bearing an ethynyl group was complex owing to the sensitive of the ethynyl group to the used oxidant.” was added.

Thank you for your kind consideration for publication in *Nature Communications*, and looking forward to your response.

Sincerely

Renhua Fan

Reviewer #1 (Remarks to the Author):

The authors have carefully revised the manuscript and completely addressed the problems raised by the referees. Therefore, I am happy to recommend publication in Nature Communications as is it.

Reviewer #2 (Remarks to the Author):

The authors have carefully revised the manuscript according to the comments of the reviewers. So, it can be accepted in current form.